# Vaginal probiotic adherence and acceptability in Rwandan women with high sexual risk participating in a pilot randomised controlled trial: a mixed-methods approach

Marijn C Verwijs [1], Stephen Agaba,[2] Marie Michele Umulisa,[2] Mireille Uwineza,[2] Adrien Nivoliez,[3] Elke Lievens,[4] Janneke H H M van de Wijgert [1,5]

For numbered affiliations see end of article.

**Correspondence to**
Professor Janneke H H M van de Wijgert;
j.vandewijgert@liverpool.ac.uk

## ABSTRACT

**Objectives** To evaluate adherence and acceptability of intermittent vaginal probiotic or antibiotic use to prevent bacterial vaginosis (BV) recurrence.

**Design** Repeated adherence and acceptability assessments using mixed methods within a pilot randomised controlled trial.

**Setting** Research clinic in Kigali, Rwanda.

**Participants** Rwandan women with high sexual risk.

**Interventions** Women diagnosed with BV and/or trichomoniasis were randomised to four groups (n=17 each) after completing metronidazole treatment: behavioural counselling only, or behavioural counselling plus 2-month intermittent use of oral metronidazole, Ecologic Femi+ (EF+) vaginal capsule or Gynophilus LP (GynLP) vaginal tablet.

**Outcome measures** Adherence and acceptability were assessed by structured face-to-face interviews, semi-structured focus group discussions and in-depth interviews, daily diaries and counting of used/unused study products in randomised women (n=68). Vaginal infection knowledge was assessed by structured face-to-face interviews in randomised women and women attending recruitment sessions (n=131).

**Results** Most women (93%) were sex workers, 99.2% were unfamiliar with BV and none had ever used probiotics. All probiotic users (n=32) reported that insertion became easier over time. Triangulated adherence data showed that 17/17 EF+ users and 13/16 GynLP users used ≥80% of required doses (Fisher's exact p=0.103). Younger age (p=0.076), asking many questions at enrolment (p=0.116), having menses (p=0.104) and reporting urogenital symptoms (p=0.103) were non-significantly associated with lower perfect adherence. Women believed that the probiotics reduced BV recurrence, but reported that partners were sometimes unsupportive of study participation. Self-reported vaginal washing practices decreased during follow-up, but sexual risk behaviours did not. Most women (12/15) with an uncircumcised steady partner discussed penile hygiene with him, but many women found this difficult, especially with male clients.

**Conclusions** High-risk women require education about vaginal infections. Vaginal probiotic acceptability and

## Strengths and limitations of this study

► We conducted this research in the context of a pilot randomised controlled trial, and statistical power was therefore limited.

► We triangulated different sources of adherence data to maximise accuracy and used a mixed-methods approach to evaluate acceptability.

► We could not directly compare experiences with, and opinions about, the two different vaginal probiotics because each woman used only one product and qualitative data depth was suboptimal.

► Social desirability bias may have affected some of the results.

► The results of this study may not be generalisable to women at lower risk of sexually transmitted or urogenital infections.

adherence were high in this cohort. Our results can be used to inform future product development and to fine-tune counselling messages in prevention programmes.

**Trial registration number** NCT02459665.

## INTRODUCTION

Bacterial vaginosis (BV) is a vaginal condition in which fastidious anaerobes such as *Gardnerella vaginalis* increase while beneficial, lactic acid–producing lactobacilli decrease.[1] Often asymptomatic, it is associated with increased risks of sexually transmitted infections (STIs) and HIV acquisition, pelvic inflammatory disease and adverse pregnancy outcomes.[2–5] Although BV is treatable with antibiotics, the risk of recurrence is high.[6 7] The prevalence of BV varies among regions and ethnic groups but is highest in sub-Saharan Africa, where it is estimated at 30%–50%.[8]

Vaginally administered probiotics containing lactobacilli are considered a promising new strategy to restore a

lactobacilli-dominated vaginal microbiota during and/or after antibiotic treatment, or to prevent BV.[9] While some probiotics have been available on the market for several years, clinical trials to support beneficial effects have only recently been initiated for most products.[10–13] Future uptake and adherence of a vaginal probiotic, once proven efficacious, is determined to a large extent by its acceptability in target populations. The acceptability, in turn, depends on factors such as characteristics of the target population, characteristics of and experiences with the product, types of sexual relationships and partner support, and community perceptions.[14 15]

We conducted a clinical trial of intermittent use of two vaginal probiotics and oral metronidazole to prevent BV recurrence in Rwandan women who had been treated for BV and/or *Trichomonas vaginalis* (TV). We used qualitative and quantitative research methods to assess adherence and acceptability with vaginal probiotic use. We triangulated various sources of adherence data to obtain adherence estimates per woman for each period of intermittent product use in between study visits, and determined correlates of adherence.

## METHODS

The pilot clinical trial took place from June 2015 to February 2016 at the Rinda Ubuzima research clinic in Kigali, Rwanda. The trial was a pilot trial with a modest sample size at the request of the funder. Women who had been successfully treated for BV/TV with a 7-day course of oral metronidazole (Tricozole; Laboratory & Allied, Nairobi, Kenya) were randomised to four intervention groups (n=17 each) to prevent BV recurrence: behavioural counselling only (controls), or behavioural counselling plus intermittent use of two different vaginal probiotics or oral metronidazole for 2 months. The behavioural counselling included counselling on safer sex, vaginal hygiene (including discouragement of intravaginal washing) and penile hygiene (ie, encouragement of cleansing the penis, including underneath the foreskin) because these behaviours are known to reduce BV recurrence risk somewhat.[6 16] We counselled all women in all randomisation groups because we considered it unethical to withhold this information from women at risk. Women were seen at screening, enrolment (product use initiation, if applicable), day 7, month 1, month 2 (product use cessation, if applicable) and month 6. Product efficacies were not known during the trial, and the efficacy results of the pilot trial are reported elsewhere.[17] Briefly, the vaginal probiotics did improve the vaginal environment (increased lactobacilli and reduced BV-associated bacteria) compared with counselling only, but not as much as oral metronidazole did.

### Study population

Women aged 18–45 at risk of HIV/STIs (defined as having had more than one sex partner and/or having been treated for an STI and/or BV in the last 12 months) were eligible for enrolment if they were confirmed HIV negative, non-pregnant, diagnosed with BV and/or TV, and cured after 7-day oral metronidazole treatment. Other clinical exclusion criteria were applied but were rare.[17] Women were recruited by study staff with the assistance of Community Mobilisers who had strong ties with local high-risk women (particularly sex workers).

### Study products and dosing

Ecologic Femi+ (EF+; Winclove Probiotics, Amsterdam, Netherlands) is a vaginal capsule containing lyophilised lactic acid–producing bacteria. EF+ was used once per day for 5 days followed by thrice weekly, for 2 months. Gynophilus LP (GynLP; Biose, Aurillac, France) is a tablet containing the *Lactobacillus rhamnosus* Lcr35 strain. The tablet disintegrates in the vagina and forms a gel that slowly releases the probiotic bacteria. GynLP was used once every 4 days for 2 months. The first dose was inserted at the clinic under direct observation of a clinician, and remaining doses were self-administered at home. Women were asked not to cleanse or insert other products into the vagina after probiotic insertion to allow the probiotics to dissolve. They were also told that they were allowed to cease probiotic use during menses, but were encouraged to continue. Intermittent metronidazole use was chosen as a positive control intervention because studies conducted in the USA and Kenya have shown a 30%–40% reduction in BV recurrence.[18 19] Metronidazole users took 500 mg generic oral metronidazole (Laboratory & Allied) twice weekly for 2 months. The rationale for selecting these study products and their dosing schedules can be found in the manuscript describing the efficacy results of the pilot trial.[17] Participants and clinicians were not blinded.

### Acceptability, adherence, behavioural and vaginal infection knowledge assessments

Acceptability was assessed at the enrolment visit prior to product use initiation and at the month 2 visit after the full 2 months of use. Adherence was assessed during the intervention period, at the day 7, month 1 and month 2 visits. Sexual and other behaviours were assessed at all study visits. Participants were interviewed face-to-face in Kinyarwanda by a trained study nurse using structured questionnaires with multiple-choice questions, questions requiring a number or date, and an adherence self-rating scale (from 0 to 10). In between visits, participants used pictorial diary cards (online supplementary material figure 1) to record daily episodes of product use, vaginal sex, condom use and vaginal practices. Those using study products returned the product packaging and unused products (if applicable) to their clinic visits, where they were counted by study staff. Any discrepancies between data sources were discussed with participants and the consensus assessments were recorded on the questionnaires. The adherence data based on the self-rating scale, the diary card and the returned product packaging were triangulated by the data analyst at the data analysis stage. In addition, 131 women were interviewed about their

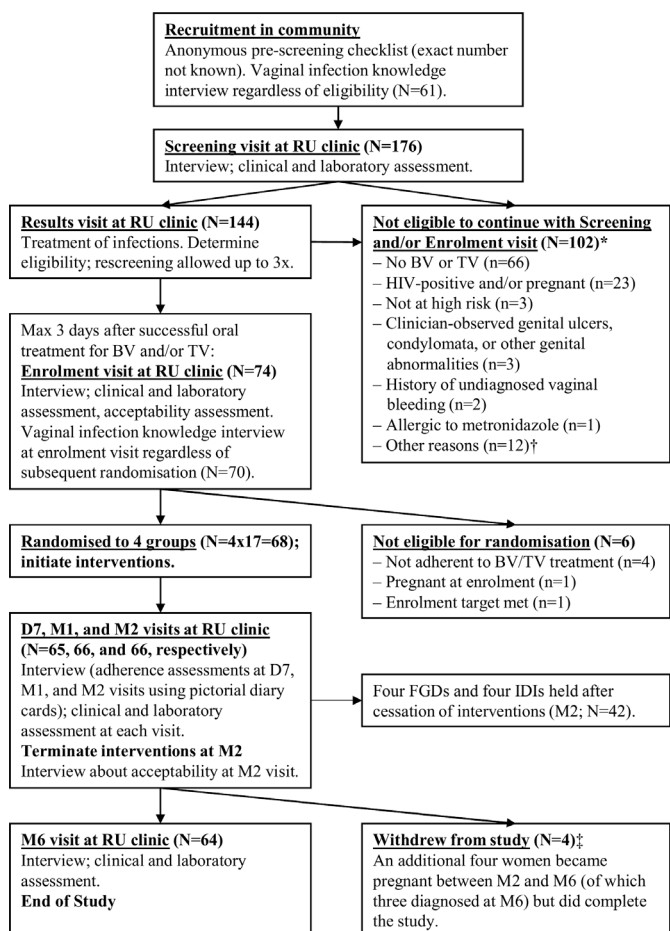

**Recruitment in community**
Anonymous pre-screening checklist (exact number not known). Vaginal infection knowledge interview regardless of eligibility (N=61).

↓

**Screening visit at RU clinic (N=176)**
Interview; clinical and laboratory assessment.

↓

**Results visit at RU clinic (N=144)**
Treatment of infections. Determine eligibility; rescreening allowed up to 3x.

→ **Not eligible to continue with Screening and/or Enrolment visit (N=102)***
– No BV or TV (n=66)
– HIV-positive and/or pregnant (n=23)
– Not at high risk (n=3)
– Clinician-observed genital ulcers, condylomata, or other genital abnormalities (n=3)
– History of undiagnosed vaginal bleeding (n=2)
– Allergic to metronidazole (n=1)
– Other reasons (n=12)†

↓

Max 3 days after successful oral treatment for BV and/or TV:
**Enrolment visit at RU clinic (N=74)**
Interview; clinical and laboratory assessment, acceptability assessment. Vaginal infection knowledge interview at enrolment visit regardless of subsequent randomisation (N=70).

↓

**Randomised to 4 groups (N=4x17=68); initiate interventions.**

→ **Not eligible for randomisation (N=6)**
– Not adherent to BV/TV treatment (n=4)
– Pregnant at enrolment (n=1)
– Enrolment target met (n=1)

↓

**D7, M1, and M2 visits at RU clinic (N=65, 66, and 66, respectively)**
Interview (adherence assessments at D7, M1, and M2 visits using pictorial diary cards); clinical and laboratory assessment at each visit.
**Terminate interventions at M2**
Interview about acceptability at M2 visit.

→ Four FGDs and four IDIs held after cessation of interventions (M2; N=42).

↓

**M6 visit at RU clinic (N=64)**
Interview; clinical and laboratory assessment.
**End of Study**

→ **Withdrew from study (N=4)‡**
An additional four women became pregnant between M2 and M6 (of which three diagnosed at M6) but did complete the study.

**Figure 1** Flowchart of the study. *Totals to 110 reasons among 102 women because there could be more than one reason per woman. †Reasons: outside of metronidazole treatment window (n=5), enrolment target already met (n=4), has a mental disorder (n=1), did not complete screening procedures and was subsequently lost to follow=up (n=1), withdrew consent during the screening visit because she thought the reimbursement was too low (n=1). ‡Reasons: moved away from Kigali (n=2), lost interest because symptoms resolved (n=1), and was verbally harassed by partner and sister about study participation (n=1). Acceptability assessments were made at enrolment and at the M2 visit. Adherence assessments were made using self-rated assessments, pictorial diary cards, and returned packaging at the D7, M1 and M2 visits (after which product use was ceased). The vaginal infection knowledge survey was held at recruitment sessions in the community and at the enrolment visit. Changes in sexual risk-taking and vaginal practices were assessed at each follow-up visit and compared with answers given during the enrolment visit. All of these themes were discussed during the eight FGDs and IDIs. BV, bacterial vaginosis; D7, day 7 visit; FGD, focus group discussion; IDI, in-depth interview; M1/2/6, month 1/2/6 visit; RU, Rinda Ubuzima; TV, *Trichomonas vaginalis*.

knowledge of vaginal infections (such as BV and STIs) using a structured questionnaire during recruitment sessions (n=61; regardless of eligibility) and at enrolment visits (n=70; this included the 68 randomised women, and two women who attended enrolment visits but turned out to be ineligible; figure 1). Women were interviewed before being counselled at study visits or before receiving information at recruitment sessions. This questionnaire contained multiple-choice and open-ended questions. Responses to the open-ended questions were categorised and discussed by two different researchers until consensus about the answer categories was reached.

Four semi-structured focus group discussions (FGDs) with 7–11 participants per group (total n=38), and semi-structured individual in-depth interviews (IDIs) with four additional participants, were held. The main themes of these FGDs and IDIs were experiences with and opinions of the study products, sexual behaviour and vaginal practices. Women randomised to the behavioural counselling only group were not approached for the FGDs and IDIs, but all other randomised participants who had completed their product use period were approached until data saturation had been achieved. The interviews were unlinked anonymous, and women used pseudonyms to enable them to talk freely despite the fact that the discussions and interviews were taped. All interviews took place between November 2015 and March 2016, were held in Kinyarwanda, recorded on tape, transcribed verbatim and translated into English. The FGD and IDI transcripts were read and discussed by three researchers (MCV, MU and JHHMvdW) at regular intervals. The Chief Investigator (JHHMvdW) decided that data saturation had been met when the fourth FGD and the fourth IDI transcript had become available in March 2016.

## Data analysis

The primary outcomes of this study were acceptability and triangulated adherence in women randomised to study product use. Secondary outcomes included vaginal infection knowledge of the target population more broadly, and behavioural changes (of the behaviours included in the counselling messages) in all randomised women. Questionnaire data were analysed using Stata V.13 (StataCorp, College Station, TX, USA). The proportion of women with ≥80%/≥90%/100% adherence in the probiotic groups were compared by Fisher's exact tests. Changes in self-reported vaginal practices and sexual behaviours over time were tested using McNemar's test for binary outcomes, and Wilcoxon's signed-rank test for continuous outcomes. To study associations of participant characteristics with triangulated adherence, we used bivariable mixed-effects models, with perfect adherence (defined as having used all doses as instructed) per interval between study visits during the intervention period as the outcome, participant identification numbers as the random effect and one participant characteristic at the time as the fixed effect. We could not determine correlates of acceptability due to limited variation in the acceptability data (reported acceptability was high throughout the trial).

The FGD and IDI transcripts were coded using NVivo V.10.0 (QSR International, Melbourne, Australia) by one single researcher (MCV). The discussions and interviews

were semi-structured, with the aforementioned themes and associated codes prepared a priori, as well as new elements that emerged from the data. The codes were derived from an acceptability framework that has been used in studies of vaginal products for contraception or HIV prevention.[14 15 20] Components of the framework include study population characteristics, product attributes, sexual encounter and relational attributes, and the contextual environment (eg, community perceptions of product use).

## Ethical statement

All participants provided written consent for study participation and separate consent for participation in FGDs/IDIs. All non-married participants aged 18–20 also required parental/guardian consent per Rwandan law at the time of the study. The participants received 3 GBP per visit (in local currency) as a reimbursement for time and transport costs. Care was taken to protect participant privacy and confidentiality. The study was sponsored by the University of Liverpool.

## Participant and public involvement

As part of the FGDs/IDIs, a subset of the enrolled participants were invited to comment on study design and experiences with the interventions. Participants were not invited to develop outcomes, interpret the results, or to contribute to the writing or editing of this document for readability or accuracy. The preliminary results of this study were discussed with 32 stakeholders during a workshop held at the Ministry of Health in Kigali, Rwanda, in December 2017. These stakeholders included representatives of the Ministry of Health, the National University of Rwanda, the National Ethics Committee, local hospitals and clinics, and local non-governmental and women's organisations.

## RESULTS

### Baseline characteristics

We screened 176 women: bacterial STI prevalence was 31.3% and BV prevalence by Gram stain Nugent scoring was 47.9%. All 68 randomised women were treated for BV and/or TV prior to randomisation and at risk of HIV/STIs, with 93.1% reporting having exchanged sex for money and/or goods in the previous month (figure 1, online supplementary material table 1). We collected 29.93 person-years of data. Four women withdrew their informed consent during the study (for reasons unrelated to study product acceptability). None were lost to follow-up.

### Adherence

Triangulated adherence was high: 17/17 (100%) of EF+ users and 13/16 (81.3%) of GynLP users used ≥80% of required doses (Fisher's exact p=0.103; table 1), and these percentages were 15/17 (88.2%) and 11/16 (68.8%) for ≥90% (p=0.225), and 10/17 (58.8%) and 8/16 (50%)

for 100% of required doses (p=0.732), respectively. In comparison, these percentages were 15/17 (88.2%), 14/17 (82.4%) and 12/17 (70.6%), respectively, for oral metronidazole users. Reported reasons of non-adherence to vaginal probiotics during face-to-face interviews were 'simply forgetting' (n=9), experiencing side effects (n=2), menses (n=2), and being away from home and having left products at home (n=1). Additional reasons for missing doses mentioned during FGDs/IDIs were being drunk (n=2) and being confused about the dosing schedule (n=2). Only one woman in the metronidazole arm reported missing doses due to experiencing side effects. Most women in FGDs reported using all doses as instructed and finding it easy to adhere, and thought that the diary cards served as a useful reminder to use the products.

## Acceptability

### Ease of use

No participants reported having heard about probiotics before study participation. After product use, all vaginal probiotic users reported feeling very comfortable with insertion and that insertion became easier over time. All but one woman reported inserting while lying down (online supplementary material table 2).

### Bodily changes and product perception

During FGDs, several women using either vaginal probiotic reported the product (partially) "coming out" during the first few uses, but that this decreased after having gained experience. Many EF+ and GynLP users reported an increase in vaginal wetness, which was considered a positive attribute by most. Some women reported increased libido. For example, one EF+ user said: "I felt a great desire to [have] sex again and again". In contrast, one metronidazole user reported a decrease in libido. Most women believed that the vaginal probiotics decreased the recurrence of symptomatic BV (our preliminary efficacy data suggest that BV incidence had in fact decreased),[17] and a few believed that they also prevented STI acquisition (the trial had insufficient statistical power to assess this).

### Support

One social harm related to vaginal probiotic use was reported: a GynLP user was verbally harassed by her partner and her sister because of her study participation, and opted to withdraw her informed consent. Reports of partner, family and community support during the FGDs/IDIs were mixed: some women reported problems with loved ones. Negative reactions from male partners were more often based on suspicions about study participation than the products themselves. One EF+ user said: "He [her partner] did not accept that. He asked me to go together with him to the clinic [a local health centre] and check if I am not HIV-positive". Another participant using metronidazole mentioned wanting to join the study to her husband, who forbade her to participate. However,

**Table 1** Adherence to study interventions

| Adherence to study products | Metronidazole (n=17) | EF+ (n=17) | GynLP (n=16) |
|---|---|---|---|
| Adherence Enr–D7, median % (IQR) | 100 (100–100) | 100 (100–100) | 100 (100–100) |
| Adherence D7–M1, median % (IQR) | 100 (100–100) | 100 (100–100) | 100 (91.7–100) |
| Adherence M1–M2, median % (IQR) | 100 (100–100) | 100 (100–100) | 100 (92.3–100) |
| Overall adherence Enr–M2, median % (IQR) | 100 (96.3–100) | 100 (100–100) | 98.3 (89.3–100) |
| Overall adherence Enr–M2 n (%) | | | |
| Perfect* | 12 (70.6) | 10 (58.8) | 8 (50.0) |
| Adherence ≥90% | 14 (82.4) | 15 (88.2) | 11 (68.8) |
| Adherence ≥80% | 15 (88.2) | 17 (100) | 13 (81.3) |
| No of times menses Enr–M2 n (%)† | | | |
| Never | 7 (41.2) | 4 (23.5) | 2 (12.5) |
| Once | 6 (35.3) | 5 (29.4) | 4 (25.0) |
| Twice | 4 (23.5) | 8 (47.1) | 10 (62.5) |
| Did not use product during menses at least once n (%) | | | |
| Yes | 4 (23.5) | 3 (17.6) | 5 (31.3) |
| NA (never had menses) | 7 (41.2) | 4 (23.5) | 2 (12.5) |
| **Self-reported reasons for non-adherence‡** | **Metronidazole** | **EF+** | **GynLP** |
| D7: Self-reported reasons why not able to use all doses as instructed n (%)§ | | | |
| Simply forgot | 0 | 2 (11.8) | 0 |
| Product had side effects | 0 | 0 | 1 (6.7)¶ |
| M1: Self-reported reasons why not able to use all doses as instructed n (%)§ | | | |
| Simply forgot | 1 (6.3) | 1 (5.9) | 1 (6.3) |
| Product had side effects | 1 (6.3)** | 0 | 1 (6.3)§§ |
| Did not like product for another reason | 1 (6.3)** | 0 | 0 |
| Other | 1 (6.3)†† | 1 (5.9)‡‡ | 2 (12.5)¶¶ |
| M2: Self-reported reasons why not able to use all doses as instructed n (%)§ | | | |
| Simply forgot | 1 (6.3) | 2 (11.8) | 3 (18.8) |
| Travelled and forgot to take product | 1 (6.3) | 0 | 1 (6.25) |
| Other | 0 | 1 (5.9)*** | 1 (6.3)††† |
| D7: Participant thinks she used product correctly most of the time n (%) | 17 (100) | 16 (94.1) | 14 (93.3) |
| M1: Participant thinks she used product correctly most of the time n (%) | 13 (86.7) | 17 (100) | 11 (68.8) |
| M2: Participant thinks she used product correctly most of the time n (%) | 15 (93.7) | 16 (94.1) | 14 (87.5) |

*Defined as 100% of the prescribed doses used at the prescribed times after nurse review of the participant's diary card and returned used packaging and unused product.
†Number of times menses in the control group: never 2 (11.8%), once 3 (17.8%), twice 11 (64.7%) and thrice 1 (5.9%).
‡Numbers of participants per randomisation group may very slightly due to loss to follow-up. Participants with ≥90% adherence not shown.
§Multiple answers possible.
¶Participant reported vulval itching and burning when passing urine.
**Participant reported mild gastritis and wanting to withdraw from the study anyway.
††Participant reported receiving oral metronidazole therapy for 7 days due to infection.
‡‡Participant reported having menses twice in 1 month; decided to use less of her product until the next study visit.
§§Participant reported genital itching, genital burning and pain during sex.
¶¶One participant reported missing the D7 study visit and therefore running out of supplies. Another participant reported not to have used the study product during menses (which she was allowed to do).
***Participant reported being drunk and therefore forgetting to take the study product.
†††Participant reported taking the study product correctly but that the product came out during menses.
D7, day 7 visit; EF+, Ecologic Femi+; Enr, enrolment visit; GynLP, Gynophilus LP; IQR, inter-quartile range; M1/2, month 1/2 visit; NA, not applicable.

she decided to join anyway: "he did not know that I was using the study product, because he had refused me to join [the] study before… I used them [the study products] without informing him". All sex workers except one stated that they had not discussed study participation with male clients.

## Worries and concerns

In the FGDs, one woman reported hearing rumours prior to enrolling that vaginal products "can damage the uterus or cause tumours in the womb". However, most participants thought that vaginal probiotics would be acceptable to Rwandan women. One GynLP user argued: "They [already] give us vaginal pills", by which she meant vaginal medications for yeast infections. We did not ask women explicitly whether they would be willing to pay for the products, but some women mentioned spontaneously in FGDs that they were concerned about future product availability and pricing. They hoped that probiotics would be distributed cheaply through the Rwandan *Mutuelle* public health insurance because they would otherwise be inaccessible to many women. One metronidazole user was concerned about a limited applicability of probiotics because BV is not diagnosed by laboratory testing in Rwanda: "They do not have adequate medical instruments to test diseases, you tell the physician how […] you feel and by guessing the disease, he gives you at least four medications, saying that you may have trichomonas, you may have syphilis, you may have gonorrhoea [she refers to syndromic management[21 22]]. At health centre-level they do not have medical equipment to test diseases, meaning that they will not know who to give that [probiotic/antibiotic maintenance therapy] medication".

## Vaginal practices and sexual risk-taking

At enrolment, 35/71 (49.3%) of the women reported to never use products inside the vagina, and at month 6, this increased to 53/65 (81.5%) (OR 5.2, 95% CI 1.96 to 17.34; table 2). During FGDs, some women understood that vaginal washing practices may increase the risk of vaginal infection, but others did not. A participant stated: "You get them [i.e., vaginal diseases] anyway… whether you wash or not". In one FGD, 10 of 11 participants (90.9%) stated having ceased vaginal practices thanks to

**Table 2**  Changes in reported vaginal cleansing practices and (sexual) behaviour between the enrolment and the M6 visit

| Self-reported sociodemographic characteristics | Enr (n=71) | M6 (n=65) | OR (95% CI)* P value* |
|---|---|---|---|
| Reports using no products inside the vagina (other than for managing menses; all participants) n (%) | 35 (49.3) | 53 (81.5) | 5.2 (1.96 to 17.34) <0.001 |
| Reports using no products inside the vagina (other than for managing menses; controls and metronidazole users only)† n (%) | 15 (44.1) | 27 (79.4) | 13.0 (1.95 to 552.5) 0.002 |
| Reports using water only n (%) | 23 (32.4) | 10 (15.4) | 0.37 (0.13 to 0.92) 0.029 |
| Reports using water and soap n (%) | 3 (4.2) | 2 (3.1) | 0.67 (0.06 to 5.82) 1.00 |
| Reports using paper, cloth or cotton wool n (%) | 9 (12.7) | 0 (0) | 0.13 (0.00 to 0.93)‡ 0.008 |
| Reports using traditional herbs, stones, powders as vaginal cleansing practice n (%) | 1 (1.4) | 1 (1.5) | 1.00 (0.01 to 78.5)‡ 1.00 |
| Mean weekly frequency of vaginal practices (95% CI) | 2.15 (0.97 to 3.34) | 0.64 (0.18 to 1.11) | NA 0.328 |
| Median no of sex partners in last month at baseline or per month during follow-up period (IQR) | 5 (3–16) | 2 (1–4) | NA <0.001 |
| Any condom use reported in past 2 weeks (Enr) or since last study visit (M6), vs no condom use reported n (%) | 64 (90.1) | 60 (92.3) | 1.67 (0.32 to 10.7) 0.727 |
| Reports exchanging sex for money/goods in past month (Enr) or since last study visit (M6) n (%) | 65 (91.5) | 58 (89.2) | 0.80 (0.16 to 3.72) 1.00 |

*McNemar's OR and p value for binary variables and Wilcoxon signed-rank test p value for continuous variables, comparing the response at M6 with the response at Enr. ORs with 95% CI were also calculated for binary pre–post data.
†n=34.
‡To enable calculation of effect measures, a zero value was replaced by 1.
CI, confidence interval; Enr, enrolment visit; IQR, inter-quartile range; M6, month 6 visit; NA, not applicable; OR, odds ratio.

the study counselling. It should be noted that in contrast to many other African populations, Rwandan women use vaginal practices to increase rather than reduce vaginal lubrication. Women mentioned the use of herbs (*umush-ishiro*), Vaseline and oils for this purpose. Self-reported sexual risk taking by face-to-face interview did not change over time, except for a significant reduction in reported numbers of sex partners in the previous month at month 6 compared with enrolment. No women in FGDs/IDIs mentioned adopting safer sex practices (such as consistent condom use) in response to the counselling messages. During face-to-face interviews at the month 2 visit, 12 of 15 women (80%) who had an uncircumcised main sex partner reported asking him to regularly clean his penis in the future (online supplementary material table 2). While most women in FGDs understood that using condoms and improved penile hygiene could reduce BV rates (as shown in[6 16]), some mentioned that they found it difficult to discuss these topics with male

partners. One participant stated that this is especially difficult being a sex worker: "a man gives you his own money and you start educating him to wash!" However, another sex worker reported refusing sex with uncircumcised clients: "you leave him, because he has a lot [of] germs". Several women reported discussing circumcision with their partners; one participant reported telling her husband: "It is better that you do circumcision because it is a good thing… you would get a chance of not contracting diseases".

### Correlates of adherence

In bivariable mixed-effects models including the probiotic groups only, no participant characteristics were significantly associated with perfect adherence (table 3). However, non-significant trends were observed. Younger age (p=0.076), asking many questions at enrolment (compared with a few questions or no questions; structurally judged by a study nurse; p=0.116), having menses

**Table 3** Participant characteristics associated with perfect adherence

| Participant characteristics | EF+ and GynLP users | | EF+, GynLP and oral metronidazole users | |
|---|---|---|---|---|
| | OR (95% CI) | P value | OR (95% CI) | P value |
| Randomisation group: GynLP vs EF+ | 0.68 (0.22 to 2.11) | 0.505 | ND | ND |
| Randomisation group | | | | |
| EF+ vs metronidazole | ND | ND | 0.53 (0.15 to 1.81) | 0.308 |
| GynLP vs metronidazole | | | 0.36 (0.11 to 1.23) | 0.103 |
| Age in years: ≥30 years vs <30 | 2.66 (0.90 to 7.82) | 0.076 | 1.60 (0.61 to 4.15) | 0.336 |
| Marital status | | | | |
| Married vs never married | 0.97 (0.14 to 6.58) | 0.976 | 1.17 (0.20 to 6.99) | 0.865 |
| Divorced vs never married | 1.18 (0.29 to 4.79) | 0.912 | 1.39 (0.42 to 4.57) | 0.586 |
| Widowed vs never married | ND | 0.991 | ND | 0.99 |
| At least some schooling vs no schooling | 1.20 (0.59 to 2.45) | 0.619 | 0.80 (0.22 to 2.95) | 0.74 |
| No of sex partners last month: five or more vs four or less. | 0.58 (0.18 to 1.83) | 0.351 | 0.49 (0.17 to 1.37) | 0.173 |
| Exchanged sex for money/goods past month | ND | 0.99 | ND | 0.986 |
| Nurse reported participant asked questions at Enr | | | | |
| Yes, many vs none | 0.19 (0.02 to 1.52) | 0.116 | 0.15 (0.02 to 1.19) | 0.072 |
| Yes, a few vs none | 0.83 (0.24 to 2.83) | 0.761 | 0.83 (0.27 to 2.57) | 0.744 |
| Had menses during study visit interval | 0.41 (0.14 to 1.20) | 0.104 | 0.26 (0.09 to 0.70) | 0.008 |
| Reported alcohol consumption during study | | | | |
| Once or twice per week vs never | 0.54 (0.14 to 2.12) | 0.373 | 0.34 (0.11 to 1.08) | 0.068 |
| More than twice per week vs never | 0.92 (0.18 to 4.81) | 0.92 | 0.81 (0.19 to 3.49) | 0.774 |
| Reported at least one urogenital symptom during study interval vs none | 0.11 (0.01 to 1.56) | 0.103 | 0.30 (0.04 to 2.16) | 0.231 |
| Reported at least one adverse event during study visit interval (excluding urogenital symptoms) vs none | 0.43 (0.10 to 1.83) | 0.253 | 0.55 (0.15 to 2.05) | 0.371 |

Sociodemographic characteristics associated with perfect adherence in bivariable mixed effects models, in the enrolment–D7, D7–M1 and M1–M2 study visit intervals.
CI, confidence interval; D7, day 7 visit; EF+, Ecologic Femi+; Enr, enrolment visit; GynLP, Gynophilus LP; M1, month 1 visit; M2, month 2 visit; ND, non-determinable; OR, odds ratio.

during the previous study interval (p=0.104) and reporting urogenital symptoms (p=0.103) were associated with a lower likelihood of perfect adherence. When including oral metronidazole users, menses was significantly associated with a lower likelihood of perfect adherence (p=0.008). There were no significant associations between randomisation group and perfect adherence.

### Vaginal infection knowledge

Almost all participants reported having heard of 'diseases of the vagina' and STIs before, but only 6/131 (4.6%) knew what bacteria were (table 4). The STIs most often spontaneously named (in numerical order) were HIV, gonorrhoea and syphilis; only one participant reported having heard of BV. After having received an explanation about what BV is, only one of 131 woman reported ever having been diagnosed with BV. Most participants could name at least one cause or potential consequence of vaginal infections. Consequences wrongfully attributed to vaginal infections were cervical cancer/tumours (5/131; 3.8%), consequences to the infant such as being born with BV or congenital malformations (6/131; 4.6%), and death (4/131; 3.1%).

### DISCUSSION

Several studies of different vaginal probiotics have been conducted, some of them in sub-Saharan Africa.[10–13] However, none reported in-depth acceptability and adherence data. Our study suggests high vaginal probiotic acceptability and adherence in high-risk Rwandan women. We found no statistically significant correlates of perfect adherence, partially due to limited statistical power, but younger age, asking many questions about product use at enrolment, current menses and reporting urogenital symptoms showed trends towards a lower likelihood of perfect adherence. Vaginal probiotics are currently unavailable on the market in most African countries, and it is important to study acceptability in different target populations to inform product development and future marketing strategies. Adherence to metronidazole was comparable with, or slightly higher than, adherence reported in previously conducted studies.[19 23]

We could not evaluate the impact of self-reported acceptability aspects on adherence because almost all women reported very high acceptability in face-to-face interviews throughout the trial. Such interviews are known to suffer from social desirability bias. However, women seemed to speak freely in the FGDs, and those data indicate that they did not have major issues with product attributes or insertion. However, some women reported difficulties due to lack of male partner support. The reported increase in vaginal wetness after probiotic insertion was not considered problematic, as lubrication during sex is preferred by most Rwandan men and women.[24] This might be different in other countries where dry sex is preferred.[25] We did find a non-significant lower adherence to GynLP compared with EF+, which

might be explained by differences in formulation: GynLP forms a gel in the vagina whereas EF+ capsules merely release lyophilised bacteria. Previous research indicated high adherence to GynLP.[26] Unfortunately, the impact of these formulation differences was insufficiently probed during the FGDs; the impact of product formulation on acceptability and adherence should be investigated in future clinical trials. Participants indicated that they found the diary cards helpful in reminding them to use their products, and we believe that self-monitoring tools might indeed be helpful in maximising adherence and therefore recommend them for use in future studies.[27]

Our data suggest that counselling was partially effective in changing behaviours that increase BV risk. While these results are encouraging, it is difficult to assess to what extent they were influenced by social desirability bias. Significantly more women reported not engaging in vaginal practices at the end of the study, and most women with uncircumcised steady male partners reported having discussed penile hygiene with them. However, many women mentioned in FGDs that they found it difficult to discuss condom use and penile hygiene with male partners, especially clients. Women reduced their sexual risks only to a limited extent during follow-up, reporting a reduction in numbers of sex partners but no differences in engaging in sex work and condom use in face-to-face interviews. We did not ask women to what extent they depended on sex work for subsistence. Women who only partially depend on sex work may find it easier to negotiate with male partners.

Two probiotics-related themes that emerged from the stakeholders consultations that had not been raised by the study participants were uncertainty about long-term side effects (women in the pilot trial used the products for only 2 months) and whether probiotic bacteria (in this case lactobacilli) could also be delivered orally instead of vaginally. We have since conducted a systematic review, which showed that long-term safety of vaginal probiotics has not yet been evaluated.[28]

Our survey with women at recruitment sessions and enrolment visits showed that high-risk Rwandan women had heard of several STIs, but were generally unaware of BV, its causes and potential consequences, and what they can do to prevent it. Experiences with HIV show that public health interventions can only succeed if healthcare professionals and the public have sufficient knowledge of causes and consequences of disease.[29–31] High-risk Rwandan women (and healthcare professionals) should therefore be educated about BV, and vaginal probiotics studies should include counselling for all participants on vaginal diseases and how to prevent them.

### Limitations

Our study had limited statistical power, and social desirability bias may have affected some of our results, as is often the case in studies of this nature. In addition, it should be noted that product efficacy, availability and cost are important determinants of acceptability, and

**Table 4** Vaginal infection knowledge

| | Recruitment (n=61) | Enrolment (n=70) | Total (n=131) |
|---|---|---|---|
| Median age (IQR) | 32 (27–35)* | 31 (27–35) | 31 (27–35) |
| Has heard of diseases of the vagina before n (%) | 60 (98.4) | 70 (100) | 130 (99.2) |
| Reports knowing what bacteria are before study n (%) | 5 (8.2) | 1 (1.4) | 6 (4.6) |
| Reports having heard about STIs before study n (%) | 61 (100) | 70 (100) | 131 (100) |
| If yes, spontaneously named, without probing† n (%) | | | |
| HIV | 58 (95.1) | 65 (92.9) | 123 (93.9) |
| Gonorrhoea | 58 (95.1) | 65 (92.9) | 123 (93.9) |
| Syphilis | 44 (72.1) | 59 (84.3) | 103 (78.7) |
| Trichomoniasis | 38 (62.3) | 48 (68.6) | 86 (65.7) |
| Hepatitis | 3 (4.9) | 3 (4.3) | 6 (4.6) |
| Yeast infection | 0 | 3 (4.3) | 3 (2.3) |
| BV | 0 | 2 (2.9) | 2 (1.5) |
| Urinary tract infection | 1 (1.6) | 1 (1.4) | 2 (1.5) |
| Chlamydia | 0 | 1 (1.4) | 1 (0.8) |
| Herpes | 0 | 1 (1.4) | 1 (0.8) |
| HPV/cervical cancer | 1 (1.6) | 0 | 1 (0.8) |
| Reports having heard about BV before this study n (%) | 1 (1.6) | 0 | 1 (0.8) |
| Spontaneously reported reasons why women get vaginal disease, without probing† n (%) | | | |
| Poor toilet hygiene | 37 (60.7) | 40 (57.1) | 77 (58.8) |
| Multiple sex partners | 28 (45.9) | 36 (51.4) | 64 (48.9) |
| After sex | 25 (41.0) | 30 (43.0) | 55 (42.0) |
| Dirty underwear | 19 (31.2) | 35 (50.0) | 54 (41.2) |
| Poor vaginal hygiene | 26 (42.6) | 22 (31.4) | 48 (36.6) |
| Poor penile hygiene of male partner(s) | 4 (6.6) | 17 (24.3) | 21 (16.0) |
| Traditional vaginal practices and washing | 3 (4.9) | 12 (17.1) | 15 (11.5) |
| New sex partner | 6 (9.8) | 3 (4.3) | 9 (6.9) |
| Use of contraception | 1 (1.6) | 3 (4.3) | 4 (3.1) |
| (Improper) use of sanitary pads or tampons | 1 (1.6) | 3 (4.3) | 4 (3.1) |
| Other | 3 (4.9)‡ | 1 (1.4)§ | 4 (3.1) |
| Cannot name any reasons | 1 (1.6) | 0 | 1 (0.8) |
| Spontaneously reported negative consequences of vaginal disease being named, without probing† n (%) | | | |
| Foul smell from the vagina | 30 (49.2) | 39 (56.5) | 69 (53.1) |
| Difficulty getting pregnant | 18 (29.5) | 33 (47.8) | 51 (39.2) |
| Miscarriage | 16 (26.2) | 33 (47.8) | 49 (37.7) |
| Abnormal vaginal discharge | 12 (19.7) | 28 (40.6) | 40 (30.8) |
| Baby born too early | 16 (26.2) | 22 (31.9) | 38 (29.2) |
| Severe infection/fever of the woman | 7 (11.5) | 7 (10.1) | 14 (10.8) |
| Infection/fever of the newborn baby | 5 (8.2) | 3 (4.4) | 8 (6.1) |
| Itching | 4 (6.6) | 4 (5.8) | 8 (6.1) |
| Other consequences to the baby: being born with BV, congenital malformations and others | 3 (4.9) | 3 (4.4) | 6 (4.6) |
| Cervical cancer or tumours | 2 (3.3) | 3 (4.4) | 5 (3.8) |
| Death | 4 (6.6) | 0 | 4 (3.1) |

Continued

**Table 4** Continued

| | Recruitment (n=61) | Enrolment (n=70) | Total (n=131) |
|---|---|---|---|
| HIV/STIs | 1 (1.6) | 3 (4.4) | 4 (3.1) |
| Pain during intercourse | 0 | 3 (4.4) | 3 (2.3) |
| Cannot name any consequence | 17 (27.9) | 19 (27.5) | 36 (27.7) |

*One missing value.
†Open-ended question. Totals may be more than 100%.
‡Participants report: "If you are infected with STIs", sharing underwear and unprotected sex.
§Participant reports: vaginal medicine.
BV, bacterial vaginosis; HPV, human papilloma virus; IQR, inter-quartile range; STI, sexually transmitted infection.

were not evaluated in our study, although preliminary efficacy results in this study were promising.[17] We could not directly compare experiences with, and opinions about, the two different vaginal probiotics because each woman used only one product and qualitative data depth was suboptimal. In the FGDs/IDIs, it was sometimes difficult to ascertain whether participants were referring to personal experiences or to wider community perceptions. Strengths of our study include the use of a mixed-methods approach and triangulated adherence data.

## CONCLUSIONS

The prevention of BV recurrence will likely have to include several components to be successful, such as improved diagnostics, treatments and prophylactic products (for example probiotics), but also improved information, education, and counselling messages targeted to at-risk women and their partners. The results of this study can be used to inform future product development and to fine-tune counselling messages in future trials.

**Author affiliations**
[1] Department of Clinical Infection, Microbiology and Immunology, Institute of Infection and Global Health, University of Liverpool, Liverpool, UK
[2] Rinda Ubuzima, Kigali, Rwanda
[3] Biose, Aurillac, France
[4] Winclove, Amsterdam, The Netherlands
[5] Julius Center for Health Sciences and Primary Care, Universitair Medisch Centrum Utrecht, Utrecht, The Netherlands

**Acknowledgements** We thank the study participants, the Rinda Ubuzima team and other colleagues in Rwanda and the UK who contributed, the funders of this study, as well as the Trial Steering Committee for trial oversight. We thank Biose and Winclove for the donation of study products.

**Contributors** JHHMvdW obtained the research funding and wrote the study protocol and data collection documents. AN, EL, SA and JHHMvdW were members of the Trial Steering Committee. SA, MU, MMU and JHHMvdW collected the primary data. MU and MMU performed the FGDs and IDIs. MCV and JHHMvdW developed the analytical approach and performed the statistical analyses. MCV and JHHMvdW wrote the manuscript. All authors commented on and approved the final manuscript.

**Funding** This work was funded by the DFID/MRC/Wellcome Trust Joint Global Health Trials Scheme as a Development Project (grant reference MR/M017443/1; grant title: "Preparing for a clinical trial of interventions to maintain normal vaginal microbiota for preventing adverse reproductive health outcomes in Africa"). Vaginal probiotics for use in the trial were donated free of charge by Winclove Probiotics (Amsterdam, The Netherlands) and Biose (formerly Probionov; Aurillac, France).

**Disclaimer** The findings and conclusions in this paper are those of the authors and do not necessarily represent the views of the authors' institutions or companies, or the funder. None of the authors were paid to write this article. The corresponding author had full access to the data and had final responsibility for the decision to submit for publication.

**Competing interests** AN is employed by Biose (owner of trial product GynLP) and EL by Winclove Probiotics BV (owner of trial product EF+). AN has financial and/or intellectual investments in competing products.

**Patient and public involvement** Patients and/or the public were involved in the design, or conduct, or reporting, or dissemination plans of this research. Refer to the Methods section for further details.

**Patient consent for publication** Not required.

**Ethics approval** The study was approved by the Rwanda National Ethics Committee and the University of Liverpool Research Ethics Subcommittee for Physical Interventions.

**Provenance and peer review** Not commissioned; externally peer reviewed.

**Data availability statement** Data are available on reasonable request. The data supporting the findings of this publication are retained by the corresponding author (JHHMvdW) and will not be made openly accessible due to privacy concerns. Fully anonymised data can be made available by written request to j.vandewijgert@liverpool.ac.uk after assurance that the intended data usage is compliant with relevant ethical approvals and privacy will be maintained.

**ORCID iDs**
Marijn C Verwijs http://orcid.org/0000-0002-7745-5148
Janneke H H M van de Wijgert http://orcid.org/0000-0003-2728-4560

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
