## [Reviewer comments · BMJ Open]

ARTICLE DETAILS

TITLE (PROVISIONAL)	Vaginal Probiotic Adherence and Acceptability in Rwandan Women with High Sexual Risk Participating in a Pilot Randomised Controlled Trial: A Mixed-Methods Approach
AUTHORS	Verwijs, Marijn; Agaba, Stephen; Umulisa, Marie; Uwineza, Mireille; Niveliez, Adrien; Lievens, Elke; van de Wijgert, Janneke H.H.M.

VERSION 1 – REVIEW

REVIEWER	Michelle Catherine Sabo University of Washington, United States
REVIEW RETURNED	02-Jul-2019

GENERAL COMMENTS	This is a well written study of acceptability and adherence of probiotics in Rwandan women at high risk for HIV acquisition. Although some groups have previously shown that intra-vaginal treatments for BV are acceptable (REF), numerous other studies have demonstrated significant variability in women's acceptance and use of intra-vaginal study products (e.g., the VOICE trial). Given this variability, I believe the authors are wise to establish the acceptability of the study interventions, and publish their experiences. With that in mind, I might recommend that the author's emphasize how important it is to consider product acceptability among individual populations in the introduction. This will help readers less familiar with the field understand the relevance of the publication. Additional minor points: - Introduction, line 75, would replace the word "transmission" with "acquisition", as I believe BV is more accurately associated with HIV acquisition in women (not transmission in general).- Line 106: It would be helpful, if possible, to include the references and a brief explanation as to why women were counseled on safe sex, vaginal washing, etc. Condomless sex and vaginal washing have been associated with BV, but the general reader may not know this.- Line 301, would also recommend including a citation to support the statement that condom use and penile cleansing reduce BV- authors could more clearly define outcomes (measured adherence with qualitative reporting?) . I don't know that I would have commented on this initially, given the descriptive and exploratory nature of the study, but both the journal and STROBE profile request a clear definition of outcomes, and I think the primary outcomes could be more clearly defined.- In the second to last row of table 2, I am confused about whether the column shows any report of condomless sex versus any report of condom use. Could this row be clarified? Because women did use the diary cards to report condomless sex, could this be
--

	reported as a continuous variable (median acts of condomless sex?) -In general, if women were reporting fewer sex partners, how were they making up that income? Does that influence the author's thoughts on whether the report of fewer partners was due to social desirability bias (third paragraph of discussion)? -For completeness, would add a sentence to the results comparing adherence in a study of periodic presumptive treatment with intravaginal metronidazole/miconazole to reduce BV prevalence in female sex workers in Kenya. This study showed a reduction in BV, however adherence was much less than that reported in this study. (McClelland et al, 2015, PMID: 25526757). - The last sentence in the figure legend for Figure 1 is grammatically incorrect.
--	---

REVIEWER	Agnes Ssali MRC/JVRI & LSHTM Uganda Research Unit Uganda
REVIEW RETURNED	15-Oct-2019

GENERAL COMMENTS	I think this is a well written manuscript, my concern however is that the abstract mentions the study of 68 participants but the results emphasise, reflect survey findings of 131 participants(as the outcome measure-which I think was for the main trial). If the authors can clarify this, then the abstract would be complete reflecting the results from the targeted 68 participants for the paper. If this is made clear then the results can easily be followed and conclusions drawn. Additional information on how the 4 IDI participants were selected would add to the qualitative description of the study and was there anything unique about the findings from these four. The methods of getting the themes from the data could be made clearer since the qualitative data presented here reflects there were specific themes drawn from the data.
--

REVIEWER	Sara Vargas The Miriam Hospital and Brown Medical School, Providence, Rhode Island, USA
REVIEW RETURNED	21-Oct-2019

GENERAL COMMENTS	This paper reports adherence and acceptability from a pilot RCT to reduce BV recurrence. I see that the authors have satisfactorily addressed comments from previous review(s). I offer the following questions and suggestions for their consideration: ABSTRACT: Results: Authors note “all probiotic users reported...”, for clarity, could put the n in parenthesis, for example, “all probiotic users (n=34) reported...” and “100% of EF+ users (n=17) and 88.2% of GynLP users (n=17)...” INTRODUCTION: The authors should provide a brief rationale for why they focused on BV but included individuals who had been treated for TV, especially since the outcome seems to be BV recurrence only. METHODS: Study products and dosing: The authors should note in the paper if the dosing schedule is standard for the study products or if they were altered in any way for this trial. Acceptability, adherence, behavioral....: To be clear, of the 68 women in the study, 68-17=51 women were eligible to take part in
---

	FGDs or IDIs – correct? And of those 51, 38 took part in FGDs, 4 in IDIs, and the rest neither? Participant and public involvement: The authors note that some enrolled participants were invited to comment on study design and experiences. How was the subset selected? How many commented? What did they say about it? Are there any themes from the stakeholder discussion that should be included in the discussion section of this paper (that are not already in there)? You may want to identify topics that emerged from that discussion, if applicable. RESULTS: Baseline characteristics: Impressive that no one was lost to follow-up. How was such excellent retention achieved? Adherence: Quite interesting that 2 people were confused about the dosing schedules. Are the dosing schedules used in this study typical for the study products? These dosing regimens did seem more on the complicated side. Could the authors provide details from interviews (if they have it) about what strategies participants implemented to maintain such high reported adherence to the dosing schedule other than the diary cards, if applicable? Worries and concerns: Interesting that participants thought it would be difficult for providers to know when to give women pro-biotics and noted that they had limited diagnostic testing. Has the research team worked with medical providers to determine their willingness and perceived feasibility of promoting pro-biotic use in Rwanda? IF so, a note about that would be useful. Vaginal practices and sexual risk taking: The authors hint at the role of self-efficacy, negotiation skills, etc. that may relate to the findings that women changed their own vaginal practices but did not report changes in behaviors that relied on male partners (noting that they found it more difficult to address those). The authors could expand up on this more in the discussion rather than just summarizing the findings. Correlates of adherence: The authors note that both younger age and having menses were associated with lower perfect adherence likelihood, but are they related or are there reasons other than age (e.g., prolonged breast feeding, certain contraceptives) that prevented menstrual periods during the study interval? How was “asking many questions at enrollment” measured – was there a standard question or set of questions for each nurse to note how many questions were asked? What this quantified or a “sense” from the interventionist? Did the study team record the types of questions that were asked? This is potentially an interesting concept. Vaginal infection knowledge: It is not entirely clear to me how many women had been diagnosed with BV, TV, or both at baseline. I wonder what the authors think of only 1 woman (of how many?) reporting “ever having been diagnosed with BV” after receiving an explanation of the infection? This is touched upon in the discussion, but it is still not clear if the women just didn’t know this SPECIFIC infection (i.e., they could not identify having had BV by the proper name) or did not know they had an infection at all. DISCUSSION: The authors note that it is important to study acceptability “to inform product development and future marketing strategies.” It would be helpful for the authors to make some specific suggestions for how to translate the results of this study into future research questions for product development and marketing. LIMITATIONS:
--	---

	Do the authors see any specific limitations or effects on the findings that may have been due to the lack of blinding? CONCLUSIONS: It would be helpful if the authors would propose a few ways that this study can be used to informed product development and fine-tune counseling messages for the readers to consider.
--	---

REVIEWER	Jianping Liu Centre for Evidence-Based Chinese Medicine, Beijing University of Chinese Medicine, China
REVIEW RETURNED	21-Nov-2019

GENERAL COMMENTS	This is an interesting and well written work. There are several minor concerns that may need to be addressed. 1. In the title, 'high risk' may need to be targeted such as high risk with infection because if you don't read the whole text, people may not understand what kind of high risk the population faced. 2. Since quite a lot of readers may not fully understand qualitative methods, thus, I would suggest to add more information about the analytic methods for qualitative data as authors indicated use of triangulation for the data analysis. 3. I am not quite clear why authors used STROBE checklist rather than CONCORD (which is a randomized trial) while former is for observational study.
--

VERSION 1 – AUTHOR RESPONSE

Reviewer: 1

Reviewer Name

Michelle Catherine Sabo

Institution and Country

University of Washington, United States

Please state any competing interests or state 'None declared':
None declared

Please leave your comments for the authors below

This is a well written study of acceptability and adherence of pro-biotics in Rwandan women at high risk for HIV acquisition. Although some groups have previously shown that intra-vaginal treatments for BV are acceptable (REF), numerous other studies have demonstrated significant variability in women's acceptance and use of intra-vaginal study products (e.g., the VOICE trial). Given this variability, I believe the authors are wise to establish the acceptability of the study interventions, and publish their experiences. With that in mind, I might recommend that the author's emphasize how important it is to consider product acceptability among individual populations in the introduction. This will help readers less familiar with the field understand the relevance of the publication.

We thank the reviewer for the kind comments. We have emphasised the importance of acceptability by revising the text as follows: "Future uptake and adherence of a vaginal probiotic, once proven efficacious, is determined to a large extent by its acceptability in target populations. The acceptability, in turn, depends on factors such as characteristics of the target population, characteristics of and experiences with the product, types of sexual relationships and partner support, and community perceptions.[14,15]"

Additional minor points:

- Introduction, line 75, would replace the word "transmission" with "acquisition", as I believe BV is more accurately associated with HIV acquisition in women (not transmission in general).

We have amended the sentence by replacing "transmission" with "acquisition".

- Line 106: It would be helpful, if possible, to include the references and a brief explanation as to why women were counseled on safe sex, vaginal washing, etc. Condomless sex and vaginal washing have been associated with BV, but the general reader may not know this.

We have added a sentence to explain this: "The behavioural counselling included counselling on safer sex, vaginal hygiene (including discouragement of intravaginal washing), and penile hygiene (i.e. encouragement of cleansing the penis, including underneath the foreskin), because these behaviours are known to reduce BV recurrence risk somewhat.[6,16] We counselled all women in all randomisation groups because we considered it unethical to withhold this information from women at risk."

- Line 301, would also recommend including a citation to support the statement that condom use and penile cleansing reduce BV

We have included a citation to support this statement.

- authors could more clearly define outcomes (measured adherence with qualitative reporting?) . I don't know that I would have commented on this initially, given the descriptive and exploratory nature of the study, but both the journal and STROBE profile request a clear definition of outcomes, and I think the primary outcomes could be more clearly defined.

We have added the primary and secondary outcomes to the statistical analysis section as follows: "The primary outcomes of this study were acceptability and triangulated adherence in women randomised to study product use. Secondary outcomes included vaginal infection knowledge of the target population more broadly, and behavioural changes (of the behaviours included in the counselling messages) in all randomised women." Detailed explanations of how these outcomes were assessed are available in other sections of the methods.

- In the second to last row of table 2, I am confused about whether the column shows any report of condomless sex versus any report of condom use. Could this row be clarified? Because women did use the diary cards to report condomless sex, could this be reported as a continuous variable (median acts of condomless sex?)

We have clarified the row label as follows: "Any condom use reported in past two weeks (Enr) or since last study visit (M6), versus no condom use reported n (%)"

We compared data at the M6 visit with Enrolment, but diary cards were distributed to participants at enrolment and we therefore do not have diary data for the enrolment visit. Instead, we used a binary variable that compares condom use (any versus none) in the two weeks prior to the enrolment visit with condom use (any versus none) in the study visit interval between M2 and M6. We hope that this sentence is now clear.

-In general, if women were reporting fewer sex partners, how were they making up that income? Does that influence the author's thoughts on whether the report of fewer partners was due to social desirability bias (third paragraph of discussion)?

This is an excellent point, but unfortunately, we did not ask women to what extent they depended on sex work for their subsistence. We have added this limitation to the discussion as follows: "We did not ask women to what extent they depended on sex work for subsistence. Women who only partially depend on sex work may find it easier to negotiate with male partners."

-For completeness, would add a sentence to the results comparing adherence in a study of periodic presumptive treatment with intravaginal metronidazole/miconazole to reduce BV prevalence in female sex workers in Kenya. This study showed a reduction in BV, however adherence was much less than that reported in this study. (McClelland et al, 2015, PMID: 25526757).

We have added a sentence about this in the discussion: "Adherence to metronidazole was comparable to, or slightly higher than, adherence reported in previously conducted studies.[19,23]"

- The last sentence in the figure legend for Figure 1 is grammatically incorrect.

We have amended the sentence: "All of these themes were discussed during the eight FGDs and IDIs."

Reviewer: 2

Reviewer Name

Agnes Ssali

Institution and Country

MRC/UVRI & LSHTM Uganda Research Unit
Uganda

Please state any competing interests or state 'None declared':
None

Please leave your comments for the authors below I think this is a well written manuscript, my concern however is that the abstract mentions the study of 68 participants but the results emphasise, reflect survey findings of 131 participants(as the outcome measure-which I think was for the main trial). If the authors can clarify this, then the abstract would be complete reflecting the results from the targeted 68 participants for the paper. If this is made clear then the results can easily be followed and conclusions drawn.

We thank the reviewer for the kind comments. We amended the abstract by adding the following sentence: "Vaginal infection knowledge was assessed by structured face-to-face interviews in randomised women and women attending recruitment sessions (n=131)."

Additional information on how the 4 IDI participants were selected would add to the qualitative description of the study and was there anything unique about the findings from these four. The methods of getting the themes from the data could be made clearer since the qualitative data presented here reflects there were specific themes drawn from the data.

All women randomised to study products who had completed the trial were approached for FGDs and IDIs, and the team stopped approaching women when data saturation was reached. Therefore, women were not selected for FGDs or IDIs based on certain characteristics. We have clarified this in the methods. As also stated in the methods, the codes (=themes) were derived from an acceptability framework that has been used in studies of vaginal products for contraception or HIV prevention (citations given in the manuscript text). Components of the framework include study population characteristics, product attributes, sexual encounter and relational attributes, and the contextual environment (e.g. community perceptions of product use).

Reviewer: 3

Reviewer Name

Sara Vargas

Institution and Country

The Miriam Hospital and Brown Medical School, Providence, Rhode Island, USA

Please state any competing interests or state 'None declared':
None declared

Please leave your comments for the authors below This paper reports adherence and acceptability from a pilot RCT to reduce BV recurrence. I see that the authors have satisfactorily addressed comments from previous review(s). I offer the following questions and suggestions for their consideration:

We thank the reviewer for the kind comments.

ABSTRACT:

Results: Authors note “all probiotic users reported...”, for clarity, could put the n in parenthesis, for example, “all probiotic users (n=34) reported...” and “100% of EF+ users (n=17) and 88.2% of GynLP users (n=17)...”

In response to this comment, we have now reported numerators, denominators, and percentages throughout the manuscript.

INTRODUCTION:

The authors should provide a brief rationale for why they focused on BV but included individuals who had been treated for TV, especially since the outcome seems to be BV recurrence only.

The rationale for this has been explained in detail in the efficacy manuscript (MedRxiv URL: <https://www.medrxiv.org/content/10.1101/19001156v1>). Briefly, BV and TV require the exact same metronidazole treatment, and women with TV almost always also have BV (which was indeed the case in our own study: only two women had TV without BV). All of these women therefore have a vulnerable vaginal microbiome and are prone to BV recurrence. This is important in the context of efficacy, but is not important in the context of the current acceptability study: all women started prophylactic study product use after completion of metronidazole treatment and had no BV or TV at the time of randomisation. We have therefore not added this explanation to this manuscript, but instead, refer the interested reader to the efficacy manuscript.

METHODS:

Study products and dosing: The authors should note in the paper if the dosing schedule is standard for the study products or if they were altered in any way for this trial.

The selection of the study products and their dosing schedules are also described in detail in the efficacy manuscript (MedRxiv URL: <https://www.medrxiv.org/content/10.1101/19001156v1>). We have added the following sentence to this manuscript: “The rationale for selecting these study products and their dosing schedules can be found in the manuscript describing the efficacy results of the pilot trial. [17]”

Acceptability, adherence, behavioral....: To be clear, of the 68 women in the study, 68-17=51 women were eligible to take part in FGDs or IDIs – correct? And of those 51, 38 took part in FGDs, 4 in IDIs, and the rest neither?

Yes, this is correct, except that 4 women randomised to a product use group did not complete the trial and were therefore not approached for participation in a FGD or an IDI. Data saturation was reached after 4 FGDs (n=38) and 4 IDIs (n=4). Therefore, the majority of women who were randomised and completed the trial were involved in either a FGD or an IDI (42 of 47 women). We have clarified this in the methods.

Participant and public involvement: The authors note that some enrolled participants were invited to comment on study design and experiences. How was the subset selected? How many commented? What did they say about it?

All participants who participated in FGDs or IDIs were invited to talk about their experiences as part of those FGDs/IDIs. We have clarified this in the manuscript: “As part of the FGDs/IDIs, a subset of the enrolled participants were invited to comment on study design and experiences with the interventions.”

Are there any themes from the stakeholder discussion that should be included in the discussion section of this paper (that are not already in there)? You may want to identify topics that emerged from that discussion, if applicable.

This is an excellent suggestion and we have added the following paragraph: “Two probiotics-related themes that emerged from the stakeholders consultations that had not been raised by the study participants were uncertainty about long-term side effects (women in the pilot trial used the products for only two months) and whether probiotic bacteria (in this case lactobacilli) could also be delivered orally instead of vaginally. We have since conducted a systematic review, which showed that long-term safety of vaginal probiotics has not yet been evaluated.[28]”

RESULTS:

Baseline characteristics: Impressive that no one was lost to follow-up. How was such excellent retention achieved?

The Rinda Ubuzima research team has a longstanding presence in the sex worker community in Kigali. The team has successfully conducted studies in this community for 14 years, and has gained the community’s trust. The team engaged high risk women who participated in previous studies (the so-called community mobilisers) in recruitment and retention efforts. Moreover, sex workers in Kigali have limited access to reproductive health services, and we provided free access to such services.

Adherence: Quite interesting that 2 people were confused about the dosing schedules. Are the dosing schedules used in this study typical for the study products? These dosing regimens did seem more on the complicated side. Could the authors provide details from interviews (if they have it) about what strategies participants implemented to maintain such high reported adherence to the dosing schedule other than the diary cards, if applicable?

For the first part of the question, see our first “Methods” response to this reviewer. The dosing schedules were indeed negotiated with the companies that provided the study products and were based on product registrations with drug regulatory agencies. We do not have indications from the FGDs/IDIs that the participants used other methods to achieve high adherence, other than the diary cards.

Worries and concerns: Interesting that participants thought it would be difficult for providers to know when to give women pro-biotics and noted that they had limited diagnostic testing. Has the research team worked with medical providers to determine their willingness and perceived feasibility of promoting pro-biotic use in Rwanda? IF so, a note about that would be useful.

The stakeholder consultations included Rwandan medical providers, and they did express concerns about long-term safety of vaginal probiotics. We have added this to the discussion section: “Two probiotics-related themes that emerged from the stakeholders consultations that had not been raised by the study participants were uncertainty about long-term side effects (women in the pilot trial used the products for only two months) and whether probiotic bacteria (in this case lactobacilli) could also be delivered orally instead of vaginally. We have since conducted a systematic review, which showed that long-term safety of vaginal probiotics has not yet been evaluated.[28]” This is a theme that spontaneously emerged during the stakeholder consultations, and we did not systematically assess this and other concerns among stakeholders. We therefore do not know if this concern is widespread or not. We also do not know if this concern about long-term safety would translate into a reluctance to promote vaginal probiotics or not.

Vaginal practices and sexual risk taking: The authors hint at the role of self-efficacy, negotiation skills, etc. that may relate to the findings that women changed their own vaginal practices but did not report changes in behaviors that relied on male partners (noting that they found it more difficult to address those). The authors could expand up on this more in the discussion rather than just summarizing the findings.

This is an excellent point. Unfortunately, we can only hypothesise about self-efficacy and negotiation skills because we did not investigate this sufficiently to be able to draw firm conclusions. We have added the sentence: “We did not ask women to what extent they depended on sex work for

subsistence. Women who only partially depend on sex work may find it easier to negotiate with male partners.”

Correlates of adherence: The authors note that both younger age and having menses were associated with lower perfect adherence likelihood, but are they related or are there reasons other than age (e.g., prolonged breast feeding, certain contraceptives) that prevented menstrual periods during the study interval?

All of the study participants were post-menarche. We therefore think that age and menses were not related.

How was “asking many questions at enrollment” measured – was there a standard question or set of questions for each nurse to note how many questions were asked? What this quantified or a “sense” from the interventionist? Did the study team record the types of questions that were asked? This is potentially an interesting concept.

The standard question on the case report form that was completed by the study nurse was as follows: [During first application of the study product] Did the participant ask questions? Answer options: Yes, many; Yes, a few; No. We have clarified this in the manuscript: “compared to a few questions or no questions; structurally judged by a study nurse”.

Vaginal infection knowledge: It is not entirely clear to me how many women had been diagnosed with BV, TV, or both at baseline. I wonder what the authors think of only 1 woman (of how many?) reporting “ever having been diagnosed with BV” after receiving an explanation of the infection? This is touched upon in the discussion, but it is still not clear if the women just didn’t know this SPECIFIC infection (i.e., they could not identify having had BV by the proper name) or did not know they had an infection at all.

Regarding BV and TV: see our earlier response. At the time of randomisation, all of the women had been successfully treated for BV and/or TV, and were infection-free. Only one woman (of the 131 interviewed) had ever heard of BV prior to the study. We have added the denominator to the text. This question was asked as a general question and is not in any way related to individual medical histories of the interviewees (which were unknown to the interviewers).

DISCUSSION:

The authors note that it is important to study acceptability “to inform product development and future marketing strategies.” It would be helpful for the authors to make some specific suggestions for how to translate the results of this study into future research questions for product development and marketing.

We believe that it is too early to make specific recommendations for vaginal probiotic development and marketing. The efficacy of currently available vaginal probiotics is modest, and long-term efficacy beyond the dosing period very low, and the first priority therefore is to improve efficacy. We recently wrote a systematic review and opinion piece about this, which we have now referenced in this paper.

LIMITATIONS:

Do the authors see any specific limitations or effects on the findings that may have been due to the lack of blinding?

Blinding is important for preventing bias when assessing efficacy, but we believe that this is not true for assessing acceptability and adherence when study product efficacy and safety are either not known or not expected to differ. In this case, we knew at the start of the study that all study products were safe, and we expected – but did not know for sure – that product efficacy would be similar between the products. This is also what the participants were told. We therefore do not consider the lack of blinding a limitation in this particular paper, but it certainly was in the context of the efficacy paper.

CONCLUSIONS:

It would be helpful if the authors would propose a few ways that this study can be used to inform product development and fine-tune counseling messages for the readers to consider.

The conclusions section is meant to 'wrap-up'. We have therefore added suggestions throughout the discussion section.

Reviewer: 4

Reviewer Name

Jianping Liu

Institution and Country

Centre for Evidence-Based Chinese Medicine, Beijing University of Chinese Medicine, China

Please state any competing interests or state 'None declared':
None declared.

Please leave your comments for the authors below This is an interesting and well written work. There are several minor concerns that may need to be addressed.

We thank the reviewer for the kind comments.

1. In the title, 'high risk' may need to be targeted such as high risk with infection because if you don't read the whole text, people may not understand what kind of high risk the population faced.

We have amended this to "with high sexual risk".

2. Since quite a lot of readers may not fully understand qualitative methods, thus, I would suggest to add more information about the analytic methods for qualitative data as authors indicated use of triangulation for the data analysis.

We have added the following explanation to the methods: "The adherence data based on the self-rating scale, the diary card, and the returned product packaging were triangulated by the data analyst at the data analysis stage." What this means is that the data analyst reviewed all available adherence data for each woman at each time point and summarised that data into one overall adherence estimate. This is a common practice in mixed methods research.

3. I am not quite clear why authors used STROBE checklist rather than CONCORD (which is a randomized trial) while former is for observational study.

The CONSORT checklist does apply to the randomised controlled trial itself, and we have indeed written the primary efficacy publication in concordance with the CONSORT guidelines (MedRxiv URL: <https://www.medrxiv.org/content/10.1101/19001156v1>). However, this paper is not about the clinical trial but about the acceptability and adherence assessments that were conducted using mixed methods both within the trial (structured questionnaires and daily diaries by randomised participants) as well as parallel to the trial (survey with women at recruitment sessions; FGDs and IDIs). We therefore felt that the STROBE checklist is more appropriate.

FORMATTING AMENDMENTS (if any)

Required amendments will be listed here; please include these changes in your revised version:

1. Figure 1 citation missing:

- The in text citation for "Figure 1" is missing in your main text of your main document file. Please amend accordingly.

We have added the citation to the third sentence of the "Baseline Characteristics" paragraph of the results.

2. Required Supplementary format:

- Please re-upload your Supplementary files in PDF format.

We have uploaded the supplementary file in pdf format.

VERSION 2 – REVIEW

REVIEWER	Agnes Ssali MRC/UVRI & LSHTM
REVIEW RETURNED	06-Feb-2020
GENERAL COMMENTS	This is a very useful paper to understand the use of probiotics in such a high risk population. Two minor comments however; the researchers do not explain in the methods section how the participants that took part in FGDs and IDIs were selected. It would be useful to upload a copies of the consent forms.
REVIEWER	Sara Vargas The Miriam Hospital and Brown Medical School, Providence, Rhode Island, USA
REVIEW RETURNED	12-Feb-2020
GENERAL COMMENTS	The authors have satisfactorily addressed my comments and appear to have satisfactorily addressed the comments from the other reviewers.
REVIEWER	Jianping Liu Beijing University of Chinese Medicine, Beijing, CHINA
REVIEW RETURNED	02-Mar-2020
GENERAL COMMENTS	The study is well justified and written. One minor comment: did the investigator ask about the participants' willingness to pay for the product after finishing the study? For future implementation, this message might be meaningful.

VERSION 2 – AUTHOR RESPONSE

Reviewer: 2

Reviewer Name

Agnes Ssali

Institution and Country

MRC/UVRI & LSHTM

Please state any competing interests or state 'None declared':
None

Please leave your comments for the authors below This is a very useful paper to understand the use of probiotics in such a high risk population.

We thank the reviewer for the kind comments.

Two minor comments however; the researchers do not explain in the methods section how the participants that took part in FGDs and IDIs were selected.

All women in the three product use randomisation groups were in principle invited for the FGDs and IDIs, although some did not participate as we checked for data saturation and did not conduct any FGDs/IDIs beyond this point. We believe we have explained and addressed this in the paragraph 'Acceptability, adherence, behavioural, and vaginal infection knowledge assessments' using the following sentence: "Women randomised to the behavioural counselling only group were not approached for the FGDs and IDIs, but all other randomised participants who had completed their product use period were approached until data saturation had been achieved."

It would be useful to upload a copies of the consent forms.

We have attached the participant information sheets and informed consent forms for Screening, Enrollment, the FGDs, and the IDIs in English, but these were subsequently translated into Kinyarwanda. The Kinyarwanda versions were approved and stamped by the Rwanda National Ethics Committee and were used in the study.

Reviewer: 3

Reviewer Name

Sara Vargas

Institution and Country

The Miriam Hospital and Brown Medical School, Providence, Rhode Island, USA

Please state any competing interests or state 'None declared':
None declared

Please leave your comments for the authors below The authors have satisfactorily addressed my comments and appear to have satisfactorily addressed the comments from the other reviewers.

We thank the reviewer for the kind comments.

Reviewer: 4

Reviewer Name

Jianping Liu

Institution and Country

Beijing University of Chinese Medicine,
Beijing, CHINA

Please state any competing interests or state 'None declared':
None declared.

Please leave your comments for the authors below

The study is well justified and written. One minor comment: did the investigator ask about the participants' willingness to pay for the product after finishing the study? For future implementation, this message might be meaningful.

We thank the reviewer for the comments. We did not structurally and explicitly ask about the participants willingness to pay for the product after finishing the study (i.e., enquire about which specific price they would be willing to pay for the products), but we did enquire whether they had any financial concerns about future product availability. We have amended the following sentence to clarify this further in the article: "We did not ask women explicitly whether they would be willing to pay for the products, but some women mentioned spontaneously in FGDs that they were concerned about future product availability and pricing. They hoped that probiotics would be distributed cheaply

through the Rwandan *Mutuelle* public health insurance because they would otherwise be inaccessible to many women.”